# NASC-seq monitors RNA synthesis in single cells

Gert-Jan Hendriks [1], Lisa A. Jung [2], Anton J.M. Larsson [1], Michael Lidschreiber [2,3], Oscar Andersson Forsman[1], Katja Lidschreiber [3], Patrick Cramer [2,3] & Rickard Sandberg [1]

Sequencing of newly synthesised RNA can monitor transcriptional dynamics with great sensitivity and high temporal resolution, but is currently restricted to populations of cells. Here, we develop new transcriptome alkylation-dependent single-cell RNA sequencing (NASC-seq), to monitor newly synthesised and pre-existing RNA simultaneously in single cells. We validate the method on pre-labelled RNA, and by demonstrating that more newly synthesised RNA was detected for genes with known high mRNA turnover. Monitoring RNA synthesis during Jurkat T-cell activation with NASC-seq reveals both rapidly up- and down-regulated genes, and that induced genes are almost exclusively detected as newly transcribed. Moreover, the newly synthesised and pre-existing transcriptomes after T-cell activation are distinct, confirming that NASC-seq simultaneously measures gene expression corresponding to two time points in single cells. Altogether, NASC-seq enables precise temporal monitoring of RNA synthesis at single-cell resolution during homoeostasis, perturbation responses and cellular differentiation.

[1] Department of Cell and Molecular Biology, Karolinska Instiutet, Biomedicum, Solnavägen 9, 171 65 Solna, Sweden. [2] Department of Biosciences and Nutrition, Karolinska Institutet, NEO, Blickagången 16, 141 52 Huddinge, Sweden. [3] Department of Molecular Biology, Max Planck Institute for Biophysical Chemistry, Am Fassberg 11, 37077 Göttingen, Germany. Correspondence and requests for materials should be addressed to P.C. (email: patrick.cramer@mpibpc.mpg.de) or to R.S. (email: rickard.sandberg@ki.se)

The ability to sequence newly synthesised RNA has led to important insights into the kinetics of RNA transcription, processing, and degradation, as well as the detection of rapid transcriptional responses to cellular stimuli or perturbations[1–5]. In most cases, these approaches have relied on the incorporation of 4-thiouridine (4sU) into newly synthesised RNA during gene transcription and subsequent biochemical separation of 4sU-labelled and unlabelled RNA. As these methods require large amounts of total RNA they have been limited to studies of average cellular responses across cell populations. Recent advances in single-cell RNA sequencing and data analysis have however revealed that responses to stimuli are not uniform across cells[6,7]. This highlights the need to monitor transcriptional dynamics in single cells to capture biological variability of processes and cellular phenotypes. Although 4sU-labelled RNA cannot be isolated from single cells in quantities that allow for its sequencing, recent cell population experiments showed that chemical modification of 4sU residues incorporated in total cellular RNA leads to T–C conversions during reverse transcription that can later be read out by sequencing[8–11]. Thus, newly synthesised RNA is separated from pre-existing RNA in silico during the computational analysis of the sequencing data.

Here, we introduce a robust single-cell method for the sequencing of newly synthesised and pre-existing RNA in single cells, that we termed NASC-seq. The method is based on 4sU labelling and alkylation of the 4sU residues[8] coupled with Smart-seq2 single-cell RNA-sequencing library construction[12]. We validate the method on K562 and Jurkat cells and demonstrate that NASC-seq can identify rapidly up-regulated or down-regulated genes during transcriptional responses. We foresee a large variety of applications that can benefit from the ability to monitor the dynamics of RNA synthesis in single cells.

## Results

### Detection of 4sU-mediated base conversions in single cells.
In this study, we present a method that enables the detection of newly synthesised and pre-existing RNA in single cells that we term NASC-seq, for new transcriptome alkylation-dependent single-cell sequencing (Fig. 1a). We first applied NASC-seq to human K562 cells that were exposed to 50 μM of 4sU for 3 h, and then sorted and lysed individually. RNA was immobilised on magnetic beads with a biotinylated oligo-dT primer[13] and then alkylated[8]. Single-cell RNA-seq libraries were constructed using a modified version of Smart-seq2[12] (see "Methods" section). Reverse transcription over alkylated 4sU residues triggers the misincorporation of guanines instead of adenosines, leading to T–C conversions that identify newly synthesised RNA in the sequenced libraries[8]. Indeed, T–C conversions were the dominating type of base conversions that we observed in 4sU-labelled single cells compared to unlabelled cells (Fig. 1b). We observed T–C conversions at a frequency just below 2%, thus at similar frequencies as conversions obtained in cell population experiments[8] (Fig. 1b). We validated that miniaturised alkylation was equally efficient as bulk alkylation (Supplementary Fig. 1a) and that 4sU-labelled control (spike-in) RNA contained high levels of T–C conversions (Supplementary Fig. 1b). Interestingly, exposing cells to higher concentrations of 4sU lowered the transcriptome complexity in libraries, observed as lowered numbers of genes detected per cell (Supplementary Fig. 1c). We decided to label with 50 μM 4sU in all subsequent experiments and reduced the labelling time for K562 cells to 1 h. To further investigate the extent to which NASC-seq correctly identifies newly synthesised RNA, we analysed genes that encode mRNAs with high, intermediate, and low turnover as determined in previous population measurements[1]. We reasoned that mRNAs with high turnover

have a larger fraction of newly synthesised RNA and this should be indicated by a higher number of T–C conversions in NASC-seq data. Indeed, comparing the number of reads with T–C conversions in genes with high (*MYC*), intermediate (*PDLIM5*), or low (*GAPDH*) RNA turnover revealed that the detected RNAs had high, intermediary, and low numbers of T–C conversions, respectively (Fig. 1c; Supplementary Fig. 1d). Thus, we conclude that labelling and subsequent sequencing of newly synthesised RNA in individual cells is feasible.

### Mixture model improves identification of new transcripts.
The analysis of T–C conversions (Fig. 1c) highlighted however that 5–10% of the total reads in unlabelled cells also contained T–C conversions. As these conversions were likely caused by sequencing and PCR errors (Fig. 1b), false positives were apparent when using T–C conversions directly as a proxy for newly synthesised RNA. To improve the separation of induced conversions from background errors, we adapted a binomial mixture model from the recently published GRAND-SLAM statistical approach[14] (see "Methods" section). We estimated the true conversion probability ($p_c$) taking in consideration the background error probability ($p_e$) that we estimated from the observed amount of natural base pair mismatches in C–T and G–A conversions. Importantly, the binomial mixture model estimated a signal-to-noise ratio ($p_c/p_e$) of ~10 in this experiment (Fig. 1d) and substantially improved the detection of newly synthesised RNA (Fig. 1e) resulting in a median of 706 genes for which new reads were detected per cell. To validate the association of detecting more newly transcribed RNA (inferred new reads) in genes of higher turnover, we next sorted all genes by their mRNA turnover[1] and generated groups of genes encoding for mRNAs with low (bottom 20%) and high turnover (top 20%). As expected, genes with high mRNA turnover had significantly higher levels of newly synthesised RNA reads than genes with lower turnover (Supplementary Fig. 1e).

### NASC-seq is robust to spurious 4sU incorporation.
The ability of NASC-seq to interrogate transcriptional dynamics at high temporal precision depends on the inhibition of transcription and further 4sU label incorporation during cell handling prior to lysis (e.g. while cells are kept on ice during FACS-mediated single-cell sorting into multi-well plates). To this end we performed a control experiment where cells were incubated with 4sU on ice for 1 h followed by FACS sorting and NASC-seq library construction. Importantly, cells only exposed to 4sU on ice for 1 h had no significant increase in T–C conversions compared to unlabelled cells (Supplementary Fig. 2). In contrast, cells labelled with 4sU for only 15 min at 37 °C had a significantly increased level of T–C conversions (Supplementary Fig. 2). We conclude that NASC-seq is appropriate for experimental schemes requiring short time periods of 4sU labelling.

### Separation of newly synthesised and pre-existing RNA.
To evaluate the sensitivity and precision of NASC-seq in the identification of newly synthesised RNA in single cells, we next studied gene expression dynamics during a transcriptional response. We labelled Jurkat T-cells with 4sU, and induced a rapid transcriptional response by simultaneous addition of phorbol 12-myristate 13-acetate (PMA) and ionomycin for 30 min as described[15] (Fig. 2a). Stimulation of Jurkat cells for 30 min strongly upregulates known response genes *EGR1* and *FOS*, while housekeeping genes *ACTB* and and *GAPDH* appear unaffected (Supplementary Fig. 3a). NASC-seq revealed a high number of T–C conversions for genes that were known to be rapidly induced upon stimulation, such as *EGR1* and *FOS*, but not for the non-induced genes *GAPDH* and *ACTB* (Fig. 2b). Based on the 10 most

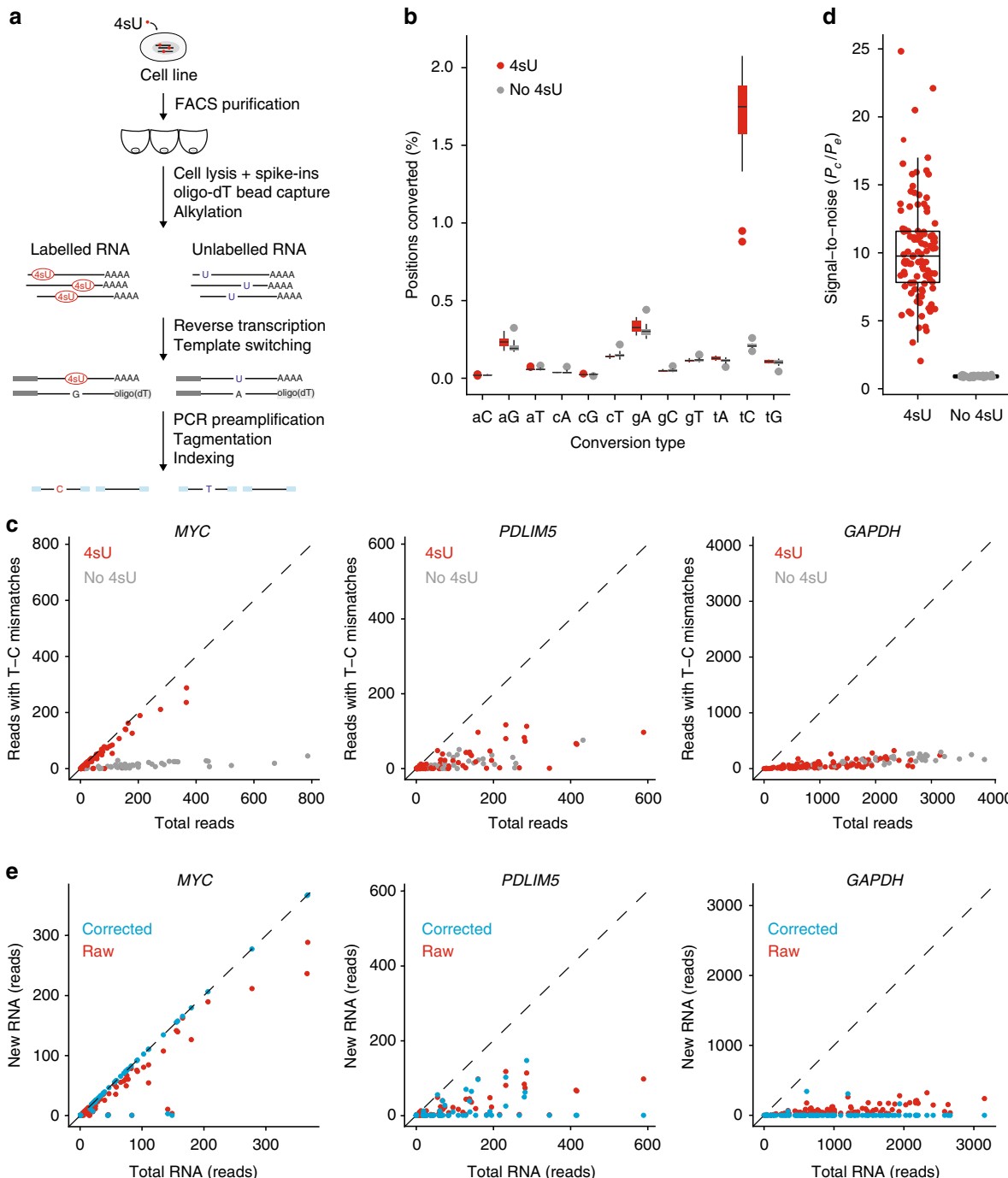

**Fig. 1** Global sequencing of newly synthesised RNA in single cells. **a** Illustration of the NASC-seq methodology. In brief, alkylation is performed on RNAs immobilised on beads for the subsequent wash before proceeding with standard PCR, tagmentation, and construction of sequencing libraries. Modified 4sU is marked in ellipses, nucleotide sequences corresponding to oligo-dT primers are shown in light grey, TSO, and ISPCR primers in dark grey, sequences added during tagmentation are indicated by blue bars. **b** Observed conversion rates in K562 cells labelled with 4sU (50μM, 3 h; red, 16 cells) or unlabelled (grey, 14 cells) on the positive strand within genes. T–C (tC) conversions are significantly ($P$-value = 1.375e−08, Mann–Whitney $U$-test, two-sided) increased in cells labelled with 4sU. Lowercase and uppercase letters indicate original and new base identities, respectively. The line in the boxplot indicates the median value, the two hinges display the first and third quartiles. The whiskers range from the hinges to the highest or lowest point that is no further than 1.5 times the interquartile range. **c** Scatter plots showing the total number of sequenced reads ($x$-axis) against the number of reads with T–C conversions ($y$-axis) for the *MYC*, *PDLIM5*, and *GAPDH* genes in 4sU labelled (50 μM, 1 h) and unlabelled cells. **d** Signal to noise estimated as $p_c$ divided by $p_e$ for K562 cells exposed to 50 μM for 1 h ($n = 106$) compared to K562 cells that were not exposed to 4sU ($n = 44$). Median, hinges, and whiskers are shown as in **b**. **e** Scatter plots showing newly synthesised (new) RNA inferred using the mixture-model (blue) or based on conversions directly (red) against total RNA (i.e. number of reads) for the same genes and 4sU-labelled cells from **c**. Source data for panels **b** and **d** are provided as a Source Data file

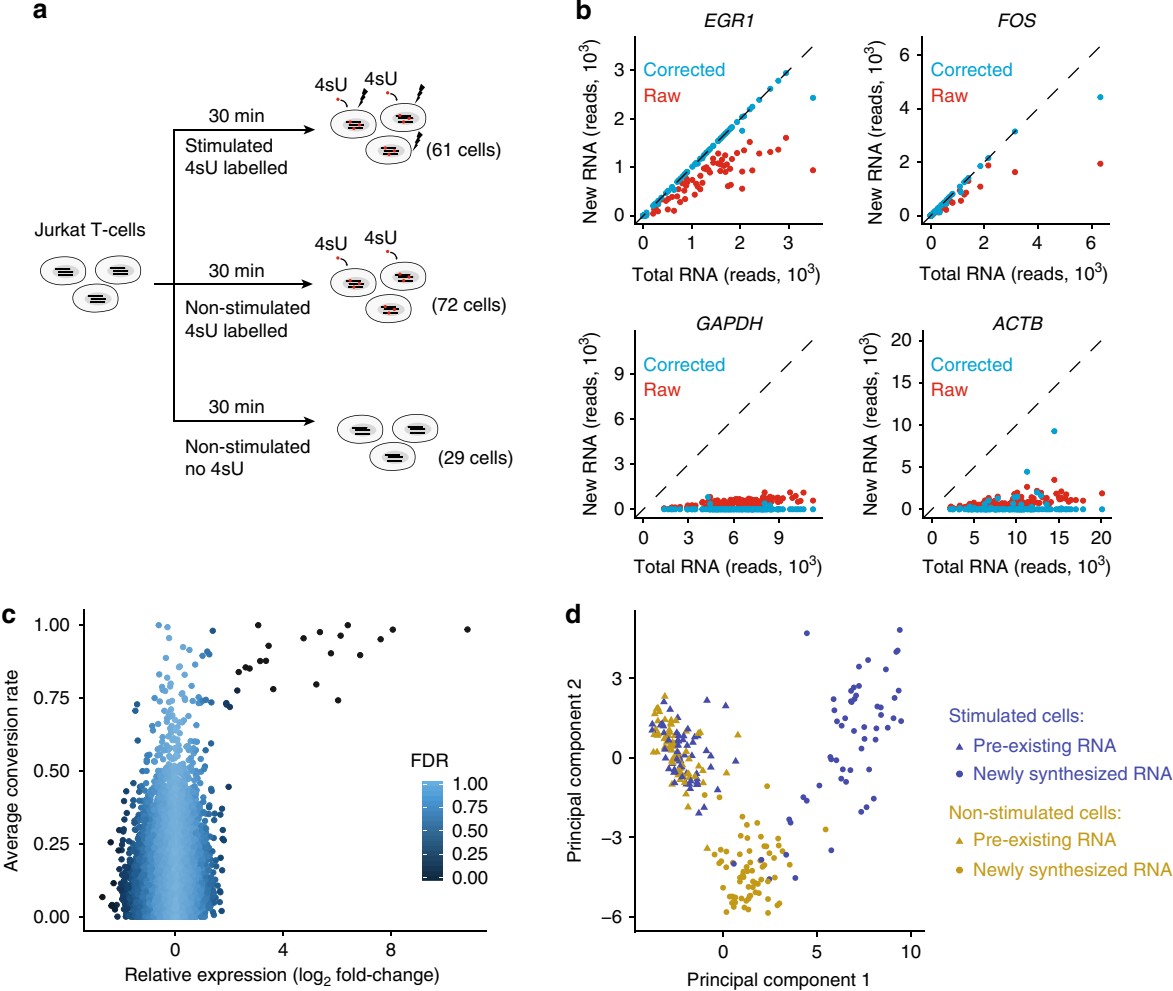

**Fig. 2** Single-cell analysis of RNA dynamics during T-cell activation. **a** Illustration of experimental design. Jurkat T-cells stimulated with PMA and ionomycin (indicated by black lightning bolts) were simultaneously exposed to 4sU (indicated by red circles) for 30 min. Unstimulated cells received only 4sU and no PMA and ionomycin. Unlabelled cells were collected that were neither stimulated nor labelled with 4sU. **b** Scatter plots of total and new RNA for two response genes (*EGR1* and *FOS*; positive controls) and two lowly turned-over genes (*GAPDH* and *ACTB*) for all 4sU-labelled cells (total $n = 133$, PMA/ionomycin stimulated $n = 61$, non-stimulated $n = 72$). **c** Scatter plots showing differential expression of genes with more than five $\pi_g$ estimates ($n = 11,022$) in Jurkat cells stimulated with PMA and ionomycin, plotted against mean conversion rate per cell. Genes were coloured according to differential expression false discovery rate (FDR) between stimulated and non-stimulated cells using ROTS[30]. **d** Two-dimensional principal component analysis (PCA) plot, using the 200 most variable genes, showing the cellular transcriptomes after separating each cell into newly synthesised and pre-existing RNA for PMA/ionomycin stimulated ($n = 61$) and non-stimulated Jurkat cells ($n = 72$)

strongly induced genes (for which we essentially only detected newly transcribed RNA), we observed on average 1.7 conversions per read (with a standard deviation of 0.66 over the genes). Application of the mixture model led to the accurate separation of newly transcribed from pre-existing RNAs, with essentially only newly transcribed RNAs for induced genes (e.g. *EGR1* and *FOS*) in contrast to lower levels of newly synthesised RNAs for housekeeping genes (e.g. *GAPDH* and *ACTB*) (Fig. 2b). Over-expressed genes in stimulated Jurkat cells, compared to non-stimulated cells, generally showed high average conversion rates, illustrating the sensitivity of NASC-seq to detect changes in expression kinetics (Fig. 2c). To further benchmark the ability of NASC-seq to detect transcriptionally induced genes upon T-cell stimulation, we selected genes that were significantly up- regulated and down-regulated, respectively, under the same conditions in bulk transient transcriptome sequencing (TT-seq) measurements and that were not detected by standard RNA-seq. Indeed, these groups of up-regulated and down-regulated genes showed a significant increase and decrease, respectively, in their

NASC-seq signal (Supplementary Fig. 3b). Altogether, NASC-seq detected transcriptionally active T-cell genes that are known to be induced under stimulating conditions[16].

**Transcriptome-wide separation of new and pre-existing RNA.** In order to investigate the ability of NASC-seq to separate newly synthesised from pre-existing RNA transcriptome-wide, we subjected total, new and pre-existing transcriptomes from T-cell activation NASC-seq data to principal component analysis (PCA). PCA could indeed separate newly synthesised RNAs in stimulated cells from those in non-stimulated cells (Fig. 2d, principal component 1, PC1). This separation was much less pronounced when total RNA measurements were used (Supplementary Fig. 3c). As expected, the separation of stimulated and non-stimulated cells was, in part, driven by known T-cell response genes, such as *EGR1* and *FOS* (Supplementary Fig. 3d and e). Also, pre-existing RNAs did not separate in this analysis, as expected (Fig. 2d). Together these analyses show that

NASC-seq can effectively measure the transcriptome at two time points per cell and it is therefore very well suited to monitor rapid changes in transcription activity in single cells.

**Differential expression in newly synthesised transcripts**. To characterise the ability of NASC-seq to resolve transcriptional dynamics, we simultaneously 4sU-labelled and stimulated Jurkat cells with PMA and ionomycin for 15 or 60 min (to complement the 30-min time point). As expected, raw conversion rates and signal-to-noise levels increased with labelling time (Supplementary Fig. 4a–f). Although the 15 min 4sU-labelled cells suffered from relatively unreliable conversion inferences (low $p_c$ estimates and higher numbers of poor $\pi_g$ estimates; see "Methods" section), they nevertheless contained a significant ($p = 6.03e{-}11$, Mann–Whitney $U$-test) median signal-to-noise ratio of 1.5 (Supplementary Fig. 4d). We identified genes with differential expression based on reads corresponding to newly transcribed RNA ('new' reads). The most strongly upregulated genes were detected both among reads from newly synthesised transcripts and reads from total RNA (Fig. 3a–c). Interestingly, an additional set of genes became upregulated during the response and was only identified as differentially expressed in newly synthesised RNAs (e.g. *EIF1* and *PTMA*) (Fig. 3b and c).

Moreover, several mitochondrial genes were only and reproducibly detected as significantly downregulated at the resolution of newly synthesised transcripts (Fig. 3a–c). The inability to detect their differential expression at the total RNA level could be due to their high average expression that we speculate could mask their transcriptional response. Importantly, the mitochondrial genes were also detected as downregulated using TT-seq (Fig. 3d–f), although with consistently lower fold-changes than those obtained using NASC-seq (Fig. 3d–f). We conclude that NASC-seq can be used to identify rapid changes in gene expression dynamics in single cells in response to perturbations, even after short labelling times.

**NASC-seq measures global RNA replacement in single cells**. Finally, we investigated the level of transcriptome-wide RNA turnover in Jurkat cells using NASC-seq data. On average, 6.5% of the RNA in cells exposed to 4sU-labelling for 30 min were newly synthesised, whereas 10.8% of the RNA in cellular transcriptomes were transcribed during 60-min labelling periods (Fig. 4a). On the individual gene-level, we observed a large variation with a median of 10.2% and 16.5% new RNA, respectively, after 30 and 60 min 4sU labelling in Jurkat T-cells (Fig. 4b).

## Discussion

In this study, we developed a method (NASC-seq) for the simultaneous quantification of newly synthesised and pre-existing RNA in single cells. NASC-seq is based on RNA labelling with 4sU, RNA modification by alkylation[8] and single-cell RNA-seq library construction using Smart-seq2[12]. We validate NASC-seq by comparison with RNA-labelling data in cell populations. We show that NASC-seq can separate newly synthesised from pre-existing RNA in single human cells, and that it can also monitor up-regulation and down-regulation of transcription during a rapid cellular response. We show that NASC-seq with as short as 15 min of 4sU labelling can detect rapidly changing gene expression, emphasising the sensitivity and robustness of NASC-seq.

Since we utilised biotinylated oligo-dT primers to immobilise and wash the RNA after alkylation, it is plausible that NASC-seq is directly compatible with simultaneous single-cell DNA-sequencing[13], scATAC-seq[17], and scNMT-seq[18]. We speculate that future combinations of these methods could expand the available toolset to allow for simultaneous measurements of transcription dynamics combined with DNA accessibility and CpG methylation status in single cells. Moreover, the NASC-seq protocol does not require any dedicated liquid handling or automation hardware and can therefore be performed in any molecular biology laboratory. A detailed step-by-step protocol for NASC-seq is provided through the protocols.io open access protocol repository[19].

The current implementation of NASC-seq leaves room for further improvements. The incorporation of unique molecular identifiers (UMIs) in the single-cell RNA-sequencing library construction would enable the direct counting of newly synthesised and pre-existing RNA, that could mitigate current biases. Short labelling times make it difficult to reliably estimate the fraction of newly transcribed RNA per gene ($\pi_g$), which can cause problems for downstream analyses. It is possible that using increased 4sU concentrations could benefit short labelling times, even though increased 4sU concentrations seemed to lower RNA yields from the cells. Despite obtaining more unreliable estimates in our 15-min labelling experiment, noise-resistant downstream analyses can still provide valuable insights (Fig. 3a). The computational analysis of NASC-seq data is challenging and CPU-intensive. Although the GRAND-SLAM statistical inference method greatly improved the separation of newly synthesised from pre-existing RNA, it currently requires computing resources that go beyond those available to most researchers. To this end, we have constructed an Amazon machine image complete with our computational pipeline to facilitate the spread of this computational analysis strategy (https://github.com/sandberg-lab/NASC-seq).

The bursting nature of gene expression results in relatively heterogeneous gene expression in a population of seemingly homogeneous cells[20,21] that also manifests as frequent mono-allelic gene expression observations[22]. By performing NASC-seq with short 4sU labelling and separating the transcription from each allele based on transcribed genetic variation[21], we will be able to directly assay individual bursts of gene expression occurring in single cells. This will be important to unambiguously validate time-scales of individual bursts inferred from steady-state cells[21]. As we have previously demonstrated that the two alleles of the same autosomal gene are typically independently transcribed[21,22] (except for genes that are imprinted or subject to allelic exclusion[23]), correlations between genes in general are not expected. However, NASC-seq could inform on any dependencies among closely related genes (e.g. those within topological associated domains) although that would likely require NASC-seq data with allelic resolution from thousands of homogeneous cells.

Recent studies showed that information on RNA synthesis can be obtained from the detection of intronic sequences in single cells[24,25], which can further inform on future cell fate[24]. NASC-seq expands on these possibilities by directly measuring newly synthesised RNA from periods of 4sU exposure without being restricted to genes for which intronic RNA reads are detected or limited in time by the kinetics of pre-mRNA splicing. We note that by measuring new and pre-existing RNA separately, we effectively measure the transcriptomes in a single cell at two distinct timepoints. Although the measure of new RNA is a direct readout of recent transcription, the measure of pre-existing RNAs contain a subset of the cellular transcriptome available before labelling, as these RNAs would be continuously turned over during the labelling time. Still, we believe the simultaneous measure of two cellular transcriptomes per cell will add a temporal dimension to studies of transcriptional kinetics[20,21]. Thus NASC-seq is ideally suited to monitor changes in transcription activity with high sensitivity and temporal resolution during cell differentiation, tissue engineering, and organism development.

## Methods

**Generation of RNA spike-ins**. Synthetic RNA spike-ins (from custom-made DNA sequences corresponding to ERCC-00043, ERCC-00170, ERCC-00136, ERCC-

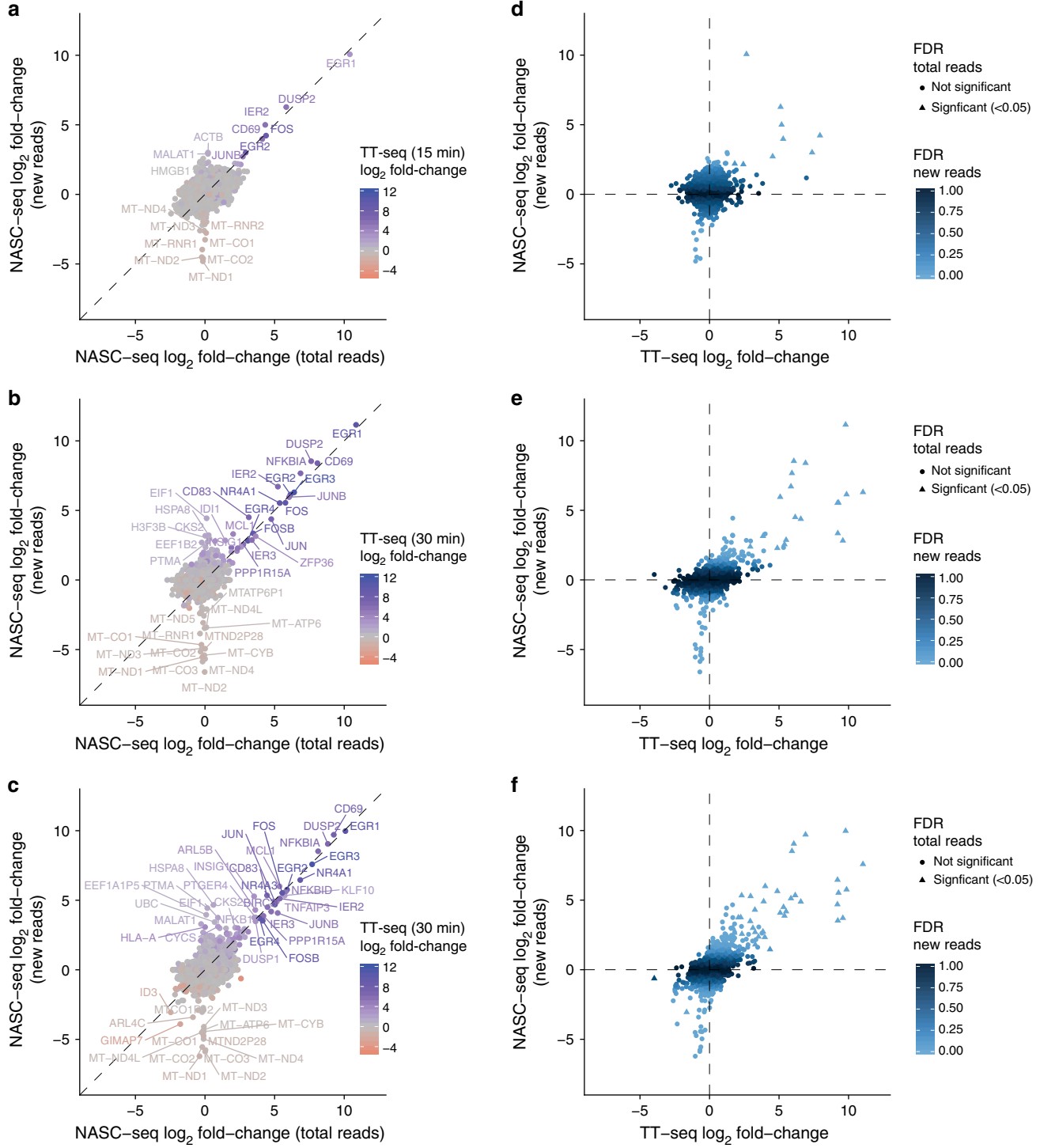

00145, ERCC-00092, and ERCC-00002, Ambion) were generated as in ref. [1]. Briefly, three 4-thiouridin (4sU)-labelled and three unlabelled RNA spike-ins were in vitro transcribed using the MEGAscript T7 Transcription Kit (Invitrogen) according to the manufacturer's instructions. Except, in vitro transcription of 4sU-labelled spike-ins was carried out substituting 10% of UTP with 4-thio-UTP (Jena Bioscience). Spike-ins were purified individually using RNAClean XP beads (Beckman Coulter) according to the manufacturer's instructions and quantified using the Qubit Fluorometer (Invitrogen). Quality was assessed using the Bioanalyzer 2100 (Agilent) and a final spike-in pool was generated containing equal amounts of all six synthetic spike-in RNAs.

**Cell culture and stimulation.** Cells were grown in RPMI 1640 medium (Gibco) supplemented with 10% foetal bovine serum (Sigma) and 1% Penicillin/Streptomycin (HyClone) at 37 °C under 5% $CO_2$. Jurkat cells (E6.1 clone) were acquired

from ATCC (TIB-152), K562 cells from DSMZ (ACC-10). K562 cells were authenticated at the DSMZ Identification Service according to standards for STR profiling (ASN-0002). Jurkat cells were purchased from ATCC shortly before use and grown at low passage numbers. Cells were routinely tested for mycoplasma contamination (MycoAlert, Lonza). On the day of the experiment, Jurkat cells were seeded to a density of $1*10^6$ cells/ml and stimulated with 50 ng/ml phorbol 12-myristate 13-acetate (PMA) and 1 μM ionomycin (Sigma) for 30 min at 37 °C under 5% $CO_2$.

**Bulk RNA alkylation.** K562 cells were labelled with 500 μM 4sU (Sigma) at 37 °C under 5% $CO_2$ for 3 h. Total RNA was isolated using TRIzol (Life Technologies) according to the manufacturer's instructions. RNA spike-in mix was added during RNA isolation. For bulk alkylation, 5 μg of RNA was alkylated as described in Herzog et al.[8]. Briefly, total RNA resuspended in 1mM DTT was treated with

**Fig. 3** NASC-seq identifies rapid transcriptional changes during T-cell stimulation. **a** Scatter plot showing differential gene expression (fold-change, log₂) in newly transcribed RNAs (y-axis) and total transcriptomes (new+pre-existing RNA, x-axis) for Jurkat cells stimulated with PMA and ionomycin for 15 min (59 cells) compared to unstimulated Jurkat cells (59 cells). Colour indicates differential expression (fold-change, log₂) from TT-seq analysis of 15 min-stimulated Jurkat cells to unstimulated Jurkat cells. Names are printed for genes with absolute log₂-fold-changes larger than 2.5. NASC-seq differential expression analysis was performed using ROTS[30] for genes expressed in at least five cells. TT-seq differential expression was performed using DESeq2[27]. The log₂-fold-changes for newly transcribed genes for which no new reads are detected were set to 0 to avoid losing data points. **b** As in **a** for 30 min-stimulated Jurkat cells (61 cells) and unstimulated cells (72 cells), coloured by differential expression from TT-seq of 30 min-stimulated Jurkat to unstimulated cells. Names are printed for genes with absolute log₂-fold-changes larger than 3. **c** As in **a** for 60 min- stimulated Jurkat cells (56 cells) and unstimulated cells (68 cells), coloured by differential expression from TT-seq of 30 min-stimulated Jurkat to unstimulated cells. Names are printed for genes with absolute log₂-fold-changes larger than 3.5. **d** Scatter plot showing differential gene expression (fold-change, log₂) for Jurkats cells stimulated for 15 min and analysed by TT-seq (x-axis) or newly transcribed RNA in NASC-seq (y-axis). Shape indicates false discovery rate (FDR) lower than 0.05 for differential expression of NASC-seq total reads. Colour indicates the FDR for differential expression of newly transcribed RNA. **e** As in **d** for 30 min-stimulated Jurkat cells. **f** As in **d** for 30 min-stimulated Jurkat cells (TT-seq) against 60 min-stimulated Jurkat cells (NASC-seq)

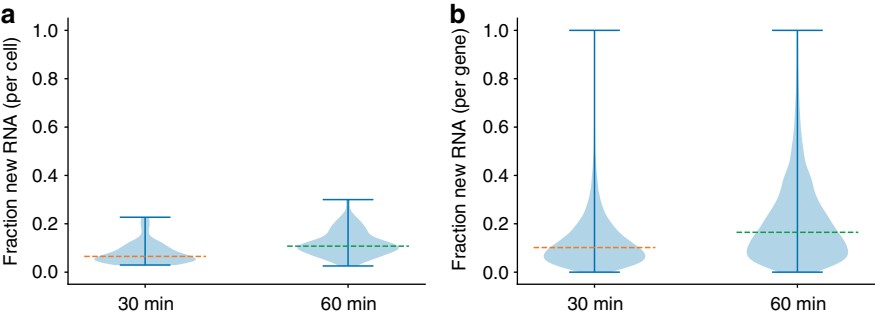

**Fig. 4** NASC-seq measures global RNA replacement rates in single cells. **a** Fraction of reads originating from newly transcribed RNA for each cell for 30 and 60-min 4sU labelling of Jurkat cells. Yellow and green lines indicate the medians, 0.065 and 0.108, respectively. The hinges denote the extreme values in their respective directions. Data is shown for 133 and 125 cells for the 30 and 60-min 4sU labelling, respectively. Stimulated and non-stimulated cells are plotted together, since the effect of stimulation does not substantially affect the global fraction of newly transcribed reads. **b** Fraction of reads originating from newly transcribed RNA per gene for 30 and 60 min labelling of Jurkat cells. Yellow and green lines indicate the medians, 0.102 and 0.165, respectively. The hinges show the extreme values as in panel **a**. Data is shown for 133 and 125 cells and 9447 and 8714 genes for the 30 and 60 min 4sU labelling, respectively. Stimulated and non-stimulated cells are plotted together, since the effect of stimulation does not substantially affect the overall distribution of fractions of new RNA per cell or gene. Source data are provided as a Source Data file

iodoacetamide (IAA) at 50 °C for 15 min in a final alkylation reaction of 50 mM sodium phosphate buffer pH 8.0, 10 mM IAA, and 50% DMSO. The reaction was quenched by adding DTT (20 mM final concentration) and RNA was purified using isopropanol precipitation. RNA was analysed using a Bioanalyzer 2100 system (Agilent). Low-input alkylation was performed as described below (NASC-seq).

**TT-seq.** Two TT-seq experiments were performed in biological duplicates, as described previously[1]. Briefly, Jurkat cells were treated for 15 or 30 min with solvent control (DMSO, Sigma) or with PMA and ionomycin. During the last 5 min of each time point cells were labelled in media with 500 µM 4sU (Sigma) at 37 °C under 5% CO₂. Cells were harvested by centrifugation at 1400×g for 2 min. Total RNA was extracted using TRIzol (Life Technologies) according to the manufacturer's instructions under the addition of spike-ins. RNAs were sonicated using in a Bioruptor Plus instrument (Diagenode). 4sU-labelled RNA was purified from 300 µg total fragmented RNA. Separation of labelled RNA was achieved with streptavidin beads (Miltenyi Biotec). Prior to library preparation, 4sU-labelled RNA treated with DNase (Qiagen), purified (miRNeasy Micro Kit, Qiagen), and quantified. Strand-specific libraries were prepared with the Ovation Universal RNA-Seq System (NuGEN). The size-selected and pre-amplified fragments were analysed on a Bioanalyzer 2100 (Agilent). Samples were sequenced on an Illumina NextSeq 500 instrument. Data analysis was performed essentially as in Michel et al. [15]. Briefly, paired-end 75 bp reads were mapped with STAR[26] (version 2.6.0c) to the hg38 (GRCh38) genome assembly (Human Genome Reference Consortium). Gene expression fold-changes upon T-cell stimulation for each time point were calculated using the R/Bioconductor implementation of DESeq2[27] setting lfcThreshold = 1. Differentially expressed genes were identified applying a P-value cutoff of 0.05 comparing sample to solvent control measurements. For comparison of NASC-seq data with TT-seq data, we calculated the fraction of new RNA (Figs. S1e and S2c) by taking the sum of reads from newly synthesised RNA divided by the sum of new and old reads over all cells, thereby creating an in silico bulk for better comparison with bulk TT-seq data. K562 TT-seq data was taken from Schwalb et al. [1] (GSE75792).

**NASC-seq.** Cells were labelled in medium with 4sU (Sigma), washed with cold PBS and sorted into lysis buffer (3 µl, 166 mM sodium phosphate pH 8.0, RNase

inhibitor, spike-in RNAs) in wells of PCR plates. Plates were frozen at −80 °C until use. Streptavidin beads (MyOne Dynabeads Strepavidin C1) were washed twice with buffer 1 (0.1 M NaOH, 0.05 M NaCl), twice with buffer 2 (0.1 M NaOH), and once with 2x B&W buffer (2 M NaCl, 10 mM Tris–HCl pH7.4, 1 mM EDTA) before the binding reaction (1x B&W, 50 µM oligo-dT). Beads were incubated with agitation at room temperature for 15 min and washed twice with 1x B&W buffer. Cells were lysed at 80 °C for 3 min and beads were added to the cells in a volume of 2 µl. RNA was bound during 20 min of incubation at room temperature on a thermoshaker (Eppendorf ThermoMixer C). 5 µl of alkylation mix (20 mM IAA in DMSO) was added for a final alkylation reaction of 50 mM sodium phosphate pH 8.0, 10 mM IAA, 50% DMSO. After 15 min at 50 °C, the alkylation reaction was stopped by adding STOP solution (2x Superscript II buffer, 0.3% Tween 20, 60 mM DTT), incubating for 5 min on a magnet, and removing the supernatant from the beads containing the alkylated RNA. RT and the remaining library preparation were performed according to a modified version of Smart-seq2[12]. The modifications included the removal of the inactivation step from the RT thermocycling programme, performing the RT on a thermoshaker (Eppendorf ThermoMixer C) and the use of custom barcoded primers for the indexing and amplification after Nextera tagmentation. The resulting libraries were sequenced on a NextSeq500 instrument (Illumina) using either single-end (75-cycle) or paired-end (2 × 150-cycle) sequencing strategies.

**Computational pipeline.** After sequencing, the resulting bcl files were demultiplexed to fastq files with *bcl2fastq* (Illumina). Nextera adapters were trimmed with *TrimGalore*[28] (v 0.4.5). All paired-end (2 × 150-cycle) data was then aligned to the hg38 human genome using STAR[26] (v 2.5). Early NASC-seq data was sequenced using a single-end (75-cycle) sequencing strategy and was aligned to hg19. For all paired-end data we removed duplicates using the MarkDuplicates command in *Picard* (v 2.17.6). We then annotated the gene each reads maps to using *FeatureCounts*[29] (v 1.6.2). We further annotated all mismatches to the reference genome within each read with the location of the mismatch for the T–C or A–G mismatches depending on the strand of the annotated gene (plus and negative, respectively). We identified mismatches in positions which appear in a high frequency over many cells and marked those positions as single nucleotide variants (SNV) to be ignored. We re-annotated the mismatches for each read with the SNV

positions ignored and avoided double counting in overlapping paired end reads. Cells with low numbers of mapping reads were removed; for Jurkat cell experiments, each cell was required to have 600,000 reads mapping to features, while for K562 experiments, cells were required to have 300,000 reads mapping to features.

We then estimated the probability of a given position being converted in a new read ($p_c$) based on the background probability of a mismatch due to error ($p_e$) for each cell. We estimated $p_e$ by calculating the mean fraction of C–T and G–A mismatches in the given cell, since we concluded that these mismatches agree with the T–C and A–G mismatches in unlabelled cells. To estimate $p_c$ we implemented the Expectation-Maximisation algorithm described in Jürges et al. [14]. For each gene in each cell, we estimated the proportion of new reads, $\pi_g$ using the binomial mixture model described in GRAND-SLAM[14] (see below). We only ran the estimation if the gene had a minimum of 16 reads mapped to ensure reliable estimates. If the gene had more than 1000 mapped reads, we subsampled down to 1000 reads to shorten runtime.

All code for the processing of NASC-seq data and the implementation of the binomial mixture model is available on GitHub (https://github.com/sandberg-lab/NASC-seq). To facilitate adoption of our computational approach we have prepared an Amazon machine image that is preloaded with our pipeline as well as example data (see GitHub for more details).

**Estimating the proportion newly transcribed transcripts**. To estimate the proportion of new reads for each gene, $\pi_g$, we implemented the binomial mixture model described in Jürges et al. [14]. In the mixture model, each mismatch is either due to a conversion with probability $p_c$ or an error with probability $p_e$. The probability of $y$ positions having mismatches in a read containing $n$ positions which may be converted is

$$P\left(y; p_e, p_c, n, \pi_g\right) = \left(1 - \pi_g\right)B(k, n, p_e) + \pi_g B(k, n, p_c) \tag{1}$$

where $B(k, n, p)$ is the binomial probability mass function.

We estimated $\pi_g$ by building a generative model in the STAN modelling language. We use a beta prior for $\pi_g$ with hyperparameters $\alpha$ and $\beta$

$$\pi_g \sim \text{Beta}(\alpha, \beta) \tag{2}$$

and estimate $\pi_g$ by maximising the log-likelihood

$$\sum_i \ln\left(P\left(y_i; p_c, p_e, n_i, \pi_g\right)\right) \tag{3}$$

where each index $i$ indicates a read for that gene. The hyperparameters were log-transformed and both initialised at 0, while $\pi_g$ was initialised at 0.5. We will then estimate the mean of the beta distribution, which cannot be 0 or 1 by definition. The mode is therefore more appropriate, which we can calculate by

$$\pi_{g\,\text{mode}} = \frac{\alpha - 1}{\alpha + \beta - 2} \tag{4}$$

for $\alpha$, $\beta > 1$. The mode is 1 if $\alpha > 1$, $\beta < 1$ and 0 if $\alpha < 1$, $\beta > 1$. The other possible cases do not occur in our estimation procedure.

**Reporting summary**. Further information on research design is available in the Nature Research Reporting Summary linked to this article.

## Data availability
K562 TT-seq data were from Schwalb et al.[1] (GSE75792). The RNA sequencing data and processed files that were generated and analysed during the current study are available in the GEO repository under accession code GSE128273. The source data underlying Figs. 1b, d and 4a, b and Supplementary Figs. 1a, c, e, 2, 3a, 4 are provided as a Source Data file. All other relevant data are available upon reasonable request.

## Code availability
The statistical model and code for inference of newly synthesised and pre-existing RNA is provided at GitHub (https://github.com/sandberg-lab/NASC-seq).

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

## Acknowledgements
We are grateful to Christoph Ziegenhain and the other members of the Sandberg laboratory for fruitful discussions. We thank Saskia Gressel, Kerstin Maier, Noah Mottelson, Anna Sawicka, Kristina Žumer (MPI-BPC Göttingen) and Aleksandra Krstic (FACS Sorting Facility Karolinska University Hospital Huddinge) for help. G.-J.H. was funded by EMBO long-term fellowship ALTF 1528-2016 and HFSP long-term fellowship LT000155/2017-L and supported by the Amazon Web Services cloud credits for research programme. L.A.J., M.L. and K.L. were funded by grants from the Center for Innovative Medicine (CIMED) and the Science for Life Laboratory (SciLifeLab). P.C. was funded by the Advanced Grant TRANSREGULON from the European Research Council, the Volkswagen Foundation, CIMED and SciLifeLab. R.S. was funded by the European Research Council (648842), the Swedish Research Council (2017-01062), the Knut and Alice Wallenberg's foundation (2017.0110) and the Bert L. and N. Kuggie Vallee Foundation.

## Author contributions

G.-J.H. and L.A.J. established and carried out NASC-seq experiments; K.L., G.-J.H. and R.S. designed the NASC-seq approach; G.-J.H., A.J.M.L., O.A.F. and M.L. developed computational methodology; A.J.M.L. and O.A.F. implemented the statistical inference approach; G.-J.H., A.J.M.L., M.L. and L.A.J. analysed data; L.A.J. carried out TT-seq experiments; G.-J.H., L.A.J., P.C. and R.S. wrote the manuscript with input from all authors; P.C. and R.S. supervised the study.

## Additional information

**Competing interests:** The authors declare no competing interests.

