## [Peer Review File · Nature Communications]

Reviewers' Comments:

Reviewer #1:

Remarks to the Author:

The manuscript "NASC-seq monitors RNA synthesis in single cells" by Hendriks et al. describes a novel method for profiling newly synthesized RNA by 4sU incorporation in single cell resolution. This method combines two well established technologies: single cell sequencing of full-length mRNA (SMART-seq) and detection of newly synthesized mRNA by 4sU incorporation followed by alkylation – resulting in an increase in T-C conversions in 4sU newly synthesized RNA.

This paper shows a proof of concept for a method that can be potentially used in many biological systems, demonstrating that:

- a. Newly synthesized RNA can be detected by the method.
- b. Total mRNA from a snapshot of single cells can be temporally sorted by the level of T-C conversions it contains.

Therefore, with several clarifications and additional analysis, we support the publication of this manuscript in Nature Communication.

Major points:

1. The authors do not provide any global assessment of new mRNA levels across genes, showing instead only three established genes (Figure 1c). Showing a global analysis is important in order to understand the detection limits of the methods. For example, the intermediate gene *Pdlim5*, after correction, seems to lack almost any synthesis. The authors should therefore show for how many genes they can detect mRNA synthesis – and cluster the genes accordingly, comparing that to previously published datasets
2. Due to the bursty nature of mRNA transcription, it will be interesting to see whether homogeneous cells show heterogeneous profiles of newly synthesized mRNA. In this direction, testing whether there interesting correlation patterns between co-synthesized genes can teach us about transcription regulation.

Minor points:

1. In the methods section the authors explain that they infer the C-T conversion background rates (p_e) for each gene separately from the untreated control. However, this can only be applied for genes that are expressed in the untreated sample. Therefore, the authors should explain how they derived p_e estimates for stimulated-induced genes such as *Egr1* and discuss whether their lack of expression in the unstimulated untreated control may affect their correction accuracy.
2. PCA plot in Figure 2d may suggests that transcription of highly transcribed house-keeping genes in stimulated cells is repressed (position on the PC2 axis), which can be an interesting result. Is there reduced synthesis of high-turnover genes? Additionally, the authors should show the top loadings of the principal components.
3. In Figure 2 there are no results from the non-stimulated/non-treated control. As in Figure 1c, the authors should compare new RNA rates in the three conditions.

Reviewer #2:

Remarks to the Author:

The work by Hendriks et al. introduces a new method for the quantitative detection of nascent/de-novo mRNA transcripts on single-cell level. The authors combine the recently published method of alkylation dependent labelling of ribonucleotides (Slam-seq) with the workflow for full-length transcript sequencing in single cells Smart-seq2 as developed in their own lab. The authors demonstrate the technical feasibility of detecting the alkylation dependent conversion of T to C mediated by incorporation of 4-thiouridine (4sU) into newly synthesized mRNA with the Smart-seq2 workflow followed by in-silico separation of pre-existing and de-novo synthesized mRNA transcripts.

The authors demonstrate the sensitivity of the approach and utilized the GRAND-SLAM algorithm

to improve the signal-to-noise ratio of their method which appears to be required to compensate for PCR and sequencing based introduction of errors leading to T to C replacements independent of 4sU incorporation. They applied their method to in-vitro cultured K562 cells and Jurkat cells stimulated with PMA and Ionomycin for 30 minutes to assess stimulation dependent increases of mRNA levels with NASC-seq.

The overall conclusions drawn by the authors are solid and the experimental workflow is interesting. However, several open points remain in order to assess the limitations of the technology. Furthermore, in order to warrant publication in Nature communications, this reviewer would require to see data at least from a more dynamic biological setting or to have the method applied to a biological question that can better demonstrate the usability of this method to a broader readership (see comments below). However, in the current stage, this manuscript is not yet ready for publication in Nature Communications but I'm confident that the authors will be able to address the specific points in due time.

Major Comments:

1) The authors assess the percentage of reads that are found to be converted after 4sU exposure but fail to assess the percentage of T to C conversions per gene. This question is partially addressed in Supplementary Figure 1d but no quantification is provided to get a feeling for how many conversions are to be expected per gene product and whether or not there is a sequence bias present, comparing different genes across the whole transcriptome. Referring to Supplementary Figure 1d: This is meant to show the browser plots of Myc mRNA expression for 3 single cells after treatment with 4sU or left untreated. Is the variability observed in these three 4sU treated cells on the level of the browser plot (all three cells look quite different in my opinion) attributed to biological variability or technical variation, for instance due to ineffective incorporation of 4sU? I'm afraid the spike-in approach used cannot help to address this question as the spike-in RNA is not actively transcribed in the cells and will not contain any de-novo incorporated 4sU along with the cells own mRNA.

2) One major concern regarding the experimental setup is the timing of 4sU labelling. In bulk the labelling procedure is described to happen 5 minutes ahead of each sampling time point, thus the detected de-novo transcripts are generated in these 5 minutes prior to cell lysis. It is possible to synchronize the lysis of several bulk samples with only a minimum of variation introduced. However, I do not see how such a synchronization is possible with sorted cells as, even on ice, transcription is not fully terminated and the sorting procedure especially for rare-cells in a sample can take several minutes or hours. Washing cells before sorting will for sure remove most of the 4sU present in the medium but additional labelling resulting in technical variability of the readout cannot be excluded at this stage.

3) Dynamics: The authors test NASC-seq in K562 cells and in stimulated T-cells for 30 minutes. In order to get a better feeling for the usability of the method it would be good to see the dynamics of gene induction and repression in a time-course. This experiment or a similar approach with different cell types or maybe primary samples would add the biology needed in my opinion for publication in Nature Communications.

Minor Comments:

1) Comprehensive proof reading of a native speaker would add to the readability of several parts of this manuscript.

2) Supplementary Figure 1c: the x-axis labelling is missing. The legend states that it refers to different concentrations of 4sU but it is not detailed out.

3) The results section starts with "Based on these findings" referring to publications mentioned in

the introduction. It would be advisable to find a smoother and more explicit start into this section of the manuscript.

Response to reviewers

We would like to thank both reviewers for their insightful and helpful comments, which helped us improve the manuscript. When adding additional experiments for the revision, we realized that our quality-control filter for cells was slightly too strict. In the revised manuscript, we only filter for a minimum number of reads mapping to features (see methods for details). The impact is minimal but we include a few more cells per condition in Figures 1 and 2. Please find our detailed responses to all specific comments below (*responses are in italics*).

Reviewer #1 (Remarks to the Author):

The manuscript “NASC-seq monitors RNA synthesis in single cells” by Hendriks et al. describes a novel method for profiling newly synthesized RNA by 4sU incorporation in single cell resolution. This method combines two well established technologies: single cell sequencing of full-length mRNA (SMART-seq) and detection of newly synthesized mRNA by 4sU incorporation followed by alkylation – resulting in an increase in T-C conversions in 4sU newly synthesized RNA.

This paper shows a proof of concept for a method that can be potentially used in many biological systems, demonstrating that:

- a. Newly synthesized RNA can be detected by the method.
- b. Total mRNA from a snapshot of single cells can be temporally sorted by the level of T-C conversions it contains.

Therefore, with several clarifications and additional analysis, we support the publication of this manuscript in Nature Communication.

Major points:

1. The authors do not provide any global assessment of new mRNA levels across genes, showing instead only three established genes (Figure 1c). Showing a global analysis is important in order to understand the detection limits of the methods. For example, the intermediate gene *Pdlim5*, after correction, seems to lack almost any synthesis. The authors should therefore show for how many genes they can detect mRNA synthesis – and cluster the genes accordingly, comparing that to previously published datasets.

The reviewer raises an important question relating to the detection limit of the method and a global analysis that should be included. In the revised manuscript, we have addressed both points. To specifically address the detection limit of the method, we have added two additional data timepoints obtained with Jurkat cells that were stimulated with PMA and ionomycin in the presence of 4sU for 15 and 60 minutes, respectively. We present the results in a new paragraph on pages 8 & 9: “NASC-seq identifies differentially expressed genes shortly after stimulus-response”. The observed signal-to-noise increases with labeling time (new Supplementary Fig. S4 of the revised manuscript; relevant panels pasted in below, Figure 1 for reviewer) as expected, and at 15 minutes labeling it corresponds to a fairly weak signal.

Jurkat stimulation experiments

Figure 1 for reviewer: signal-to-noise ratios for 4sU labelled and unlabelled Jurkat cells.

Still, we show that even after only 15 minutes of 4sU labeling the conversions found were sufficient to detect differentially expressed genes (based on new reads) between Jurkat cells before and after stimulation (new Fig. 3 of the revised manuscript), however care should be taken when analyzing datasets where the signal-to-noise ratio is relatively low. In the revised manuscript, we comment on the special consideration needed when analyzing low-signal data in the discussion section (Page 10, paragraph 2). Overall, differential expression analysis of newly synthesized RNA shows good agreement between NASC-seq and TT-seq data (new Fig. 3 d-f).

On page 5 we now specifically mention the median number genes for which new reads are detected per cell for the 60 minute K562 experiment (706 genes). Moreover, to provide a genomic overview of the data, we computed the fraction of new RNA identified per cell (new Fig. 4a, Figure 2 for reviewer pasted below) and per gene (new Fig. 4b, Figure 2 for reviewer pasted below) after 30 and 60 minutes labeling, and present the results in a new paragraph on page 9: “NASC-seq measures global RNA replacement rates in single cells”. As expected, the fraction of new RNA at both the gene and cell level increased with labeling time. Indeed we observed that Jurkat cells renewed 6.5 % and 10.8 % of its transcriptome after 30 and 60 minutes, respectively. The wide distribution of renewed RNA percentages for genes show the wide span of transcriptional rates for genes.

Jurkat stimulation experiments

Figure 2 for reviewer: global RNA replacement rates per cell and per gene.

Altogether, we believe the revised version has been significantly improved as we now include experiments at the border of what is possible in terms of labeling time and we provide interesting estimates of the replenishment rates of RNAs in cells.

2. Due to the bursty nature of mRNA transcription, it will be interesting to see whether homogeneous cells show heterogeneous profiles of newly synthesized mRNA. In this direction, testing whether there interesting correlation patterns between co-synthesized genes can teach us about transcription regulation.

The reviewer correctly points out that bursty transcription itself will have important consequences for how correlated gene expression patterns are across single cells and defined time intervals. We tested for coordinated expression by performing pair-wise analyses of genes with new RNA detected in more than ten K562 cells after 60-minute 4sU labelling (approximately 2,500 genes, 106 cells). After multiple testing correction, we did not find any significant coordinated expression. The general absence of any apparent coordination is in agreement with previous findings from our lab that demonstrated that the two alleles of the same gene are independently transcribed in single cells (i.e. the bursting of the two alleles are unconnected in time) (Deng et al. 2014; Larsson et al. 2019). For genes that are more closely related (e.g. located in the same topologically associated domain), it is still possible that coordinated expressions may become detectable in large datasets of traditional single cell RNAseq on homogeneous cell populations. NASC-seq is likely to improve the power to detect such coordination (if it exists), especially when performed at allelic resolution. As a control, we performed the same analysis on our Jurkat cells with and without stimulation and we identified coordinated expression between stimulus-response genes (data not shown). This is however simply a consequence of stimulation that leads to the induction of genes in a subset of cells. We have added a paragraph to the discussion relating to bursty transcription, independent allelic expression and its consequences for correlation structures in single-cell data.

Minor points:

1. In the methods section the authors explain that they infer the C-T conversion background rates (p_e) for each gene separately from the untreated control. However, this can only be applied for genes that are expressed in the untreated sample. Therefore, the authors should explain how they derived p_e estimates for stimulated-induced genes such as Egr1 and discuss whether their lack of expression in the unstimulated untreated control may affect their correction accuracy.

The 4sU-induced conversions are read as T-C and A-G conversions (depending on the strand). We calculate an estimate of the probability that a mismatch is caused by errors (p_e) as the average C-T and G-A (note the opposite direction of change compared to with 4sU) conversion rate per cell. Since these conversions are not induced by 4sU, we can estimate p_e for each cell, independent of its 4sU labelling. As Figure 1b shows, for this experiment, the average between C-T and G-A conversions approximates the T-C conversion rate in unlabelled cells. To show that this is a fair and conservative estimate for the error probability, we have added the non-strand specific conversion rate plots for all datasets used in this study (new Supplementary Fig. S4 a-c) where we implemented the GRAND-SLAM methodology.

2. PCA plot in Figure 2d may suggest that transcription of highly transcribed house-keeping genes in stimulated cells is repressed (position on the PC2 axis), which can be an interesting result. Is there reduced synthesis of high-turnover genes? Additionally, the authors should show the top loadings of the principal components.

Interestingly, as the reviewer suggested, we see repression of mitochondrial genes in stimulated cells (although they seem to contribute to PC1). This is replicated in the new datasets added to the manuscript (both 15 and 60 minutes stimulation and 4sU labelling) (new Fig. 3). In the revised manuscript, we discuss the genes of differential turnover rate in the text. As suggested by the reviewer, we have added the genes contributing most to the principal components to the manuscript (revised Supplementary Fig. S3).

3. In Figure 2 there are no results from the non-stimulated/non-treated control. As in Figure 1c, the authors should compare new RNA rates in the three conditions.

We have now added a panel showing the raw conversions (as in Fig. 1c) for the different stimulation and labelling conditions as a supplementary figure (Supplementary Fig. 3a).

Reviewer #2 (Remarks to the Author):

The work by Hendriks et al. introduces a new method for the quantitative detection of nascent/de-novo mRNA transcripts on single-cell level. The authors combine the recently published method of alkylation dependent labelling of ribonucleotides (Slam-seq) with the workflow for full-length transcript sequencing in single cells Smart-seq2 as developed in their own lab. The authors demonstrate the technical feasibility of detecting the alkylation dependent conversion of T to C mediated by incorporation of 4-thiouridine (4sU) into newly synthesized mRNA with the Smart-seq2 workflow followed by in-silico separation of pre-existing and de-novo synthesized mRNA transcripts.

The authors demonstrate the sensitivity of the approach and utilized the GRAND-SLAM algorithm to improve the signal-to-noise ratio of their method which appears to be required to compensate for PCR and sequencing based introduction of errors leading to T to C replacements independent of 4sU incorporation. They applied their method to in-vitro cultured K562 cells and Jurkat cells stimulated with PMA and Ionomycin for 30minutes to assess stimulation dependent increases of mRNA levels with NASC-seq.

The overall conclusions drawn by the authors are solid and the experimental workflow is interesting. However, several open points remain in order to assess the limitations of the technology. Furthermore, in order to warrant publication in Nature communications, this reviewer would require to see data at least from a more dynamic biological setting or to have the method applied to a biological question that can better demonstrate the usability of this method to a broader readership (see comments below). However, in the current stage, this manuscript is not yet ready for publication in Nature Communications but I'm confident that the authors will be able to address the specific points in due time.

We thank the reviewer for the careful review and good suggestions. We think we could address all concerns and the manuscript is now ready for acceptance.

Major Comments:

1) The authors assess the percentage of reads that are found to be converted after 4sU exposure but fail to assess the percentage of T to C conversions per gene. This question is partially addressed in Supplementary Figure 1d but no quantification is provided to get a feeling for how many conversions are to be expected per gene product and whether or not there is a sequence bias present, comparing different genes across the whole transcriptome. Referring to Supplementary Figure 1d: This is meant to show the browser plots of *MYC* mRNA expression for 3 single cells after treatment with 4sU or left untreated. Is the variability observed in these three 4sU treated cells on the level of the browser plot (all three cells look quite different in my opinion) attributed to biological variability or technical variation, for instance due to ineffective incorporation of 4sU? I'm afraid the spike-in approach used cannot help to address this question as the spike-in RNA is not actively transcribed in the cells and will not contain any de-novo incorporated 4sU along with the cells own mRNA.

In order to compute the numbers of conversions per read and gene, we focused on the most strongly induced genes (e.g. in the 30-minute Jurkat stimulation experiment) since for these genes we observed essentially only newly transcribed RNA (and we exclude confounding effect of pre-existing RNAs). In these genes, we detected on average 1.7 conversion per read derived from newly transcribed RNA, with a standard deviation of 0.66 across the genes (Figure 3 for reviewer).

We have included this information in the main text (page 7, paragraph 1). The incorporation of 4sU observed in SLAM-seq data was shown to be evenly distributed and to not have a substantial bias (Herzog et al., 2017). The variation between cells we observe is likely derived from both limited number of RNAs per cell which can be further exaggerated due to RNA losses during single-cell RNA-seq library construction and unequal amplification. In the revised discussion, we now address these aspects of single-cell metabolic labelling experiments (page 10, paragraph 2) and speculate that the potential addition of UMIs to the NASC-seq method could counteract amplification biases, although not reduced complexity from losses during library preparation.

Figure 3 for reviewer: Conversions per read per gene for the top 10 induced genes upon 30-minute Jurkat stimulation.

2) One major concern regarding the experimental setup is the timing of 4sU labelling. In bulk the labelling procedure is described to happen 5 minutes ahead of each sampling time point, thus the detected de-novo transcripts are generated in these 5 minutes prior to cell lysis. It is possible to synchronize the lysis of several bulk samples with only a minimum of variation introduced. However, I do not see how such a synchronization is possible with sorted cells as, even on ice, transcription is not fully terminated and the sorting procedure especially for rare-cells in a sample can take several minutes or hours. Washing cells before sorting will for sure remove most of the 4sU present in the medium but additional labelling resulting in technical variability of the readout cannot be excluded at this stage.

We in principle share the reviewers concern with respect to the possibility of 4sU incorporation while cells are being prepared, considering the relative duration of our cell preparation compared to short labeling times. To estimate the extent of 4sU incorporation in cells at 4°C, we performed new experiments where 4sU was added to cells that were subsequently kept on ice in PBS for 1

hour and present the results in a new paragraph on page 6: “NASC-seq is robust to spurious 4sU incorporation during cell handling”. We compared the conversion rates observed to those from cells labelled with 4sU for 15 minutes at 37°C in growth medium. We do not see any apparent 4sU incorporation signal in the libraries from cells exposed only to 4sU when on ice in PBS (revised Supplementary Fig. S2 and Figure 4 for reviewer below). We reason that several cellular processes are needed in addition to transcription, such as cleavage, termination and polyadenylation, before a transcript can be detected in our polyA-based scRNA-seq assay, which seem very inefficient in cells on ice. Although we can’t formally rule out that some complete transcripts may be generated while cells are on ice, the extent of such uncontrolled labeling is likely minor. We thank the reviewer for bringing up this question. The new experiments provided in the revised manuscript have significantly improved understanding of the precision of the single-cell labeling approach.

K562 4sU labelling on ice

Figure 4 for reviewer: K562 cells labelled with 4sU on ice for an hour do not show significant 4sU incorporation (P -values from Mann-Whitney U -test).

3) Dynamics: The authors test NASC-seq in K562 cells and in stimulated T-cells for 30 minutes. In order to get a better feeling for the usability of the method it would be good to see the dynamics of gene induction and repression in a time-course. This experiment or a similar approach with different cell types or maybe primary samples would add the biology needed in my opinion for publication in Nature Communications.

We have added two experiments in order to emphasize the method's capabilities and to provide new biological insights into cellular RNA turnover. First, we have added new Jurkat T-cell experiments where cells were stimulated and 4sU exposed for 15 or 60 minutes, representing a time-course of 15, 30 and 60 minutes. We present the results in a new paragraph on pages 8 & 9: "NASC-seq identifies differentially expressed genes shortly after stimulus-response". We observed a rapid induction of response genes (new Fig. 3a). Longer stimulation and labeling results in the increase in the number of differentially expressed genes detected. Interestingly, NASC-seq exclusively detects genes that are differentially expressed upon stimulation with PMA/Ionomycin, which would not be detected without the separation of new and old RNAs (see genes moving along the y-axis only in new Fig. 3a). Strikingly, many of these genes were already detected already after 15 minutes of labeling and stimulation. We argue that especially for highly expressed genes, rapid changes are challenging to detect with bulk methods and impossible to detect with transcriptome-wide single-cell methods without labelling. We show that NASC-seq is able to detect rapid down- and up-regulation of a large group of highly expressed and stable genes in response to PMA/ionomycin stimulation. We believe that the ability to detect rapid changes in gene expression adds relevance to this study for a broader readership.

Second, we provide new computational analyses that summarizes what percentage of the RNAs that are new at different time points and present the results in a new paragraph on page 9: "NASC-seq measures global RNA replacement rates in single cells". Performing these analyses on the level of genes revealed large variation in their rates of replenishing. Interestingly, performing the analyses on the whole cellular transcriptomes revealed that 6.5 and 10.8 % of the cell's RNA are renewed after 30 and 60 minutes, respectively. We believe that the additional experiments performed, together with all other improvements to the revised manuscript, demonstrate that new biological insights can be obtained with the method.

Minor Comments:

1) Comprehensive proof reading of a native speaker would add to the readability of several parts of this manuscript.

We have gone through the text again to improve the language.

2) Supplementary Figure 1c: the x-axis labelling is missing. The legend states that it refers to different concentrations of 4sU but it is not detailed out.

The x-axis label was added.

3) The results section starts with "Based on these findings" referring to publications mentioned in

the introduction. It would be advisable to find a smoother and more explicit start into this section of the manuscript.

We rewrote to provide a smoother transition.

Reviewers' Comments:

Reviewer #2:

Remarks to the Author:

I would like to thank the authors for their careful assessment of all my comments. I indeed think that no additional experiments are needed to demonstrate the usability and technical limitations of the NASC-seq technology.

Two minor points remain regarding labels in the following figures:

Figure 1a - exchange the term "FACS sorting" by i.e. "FACS purification"

Figure 3c: The label should say "60 minutes" instead of "30 minutes"

REVIEWERS' COMMENTS:

Reviewer #2 (Remarks to the Author):

I would like to thank the authors for their careful assessment of all my comments. I indeed think that no additional experiments are needed to demonstrate the usability and technical limitations of the NASC-seq technology.

Two minor points remain regarding labels in the following figures:

Figure 1a - exchange the term "FACS sorting" by i.e. "FACS purification"

We have changed the term FACS sorting to FACS purification as suggested.

Figure 3c: The label should say "60 minutes" instead of "30 minutes"

The Figure label for Fig 3c was correct in stating 30 minutes. In Figure 3, we plot the log₂ fold-change of all reads against new reads only based on NASC-seq data, and we color coded genes based on TT-seq data. For A and B, we have the corresponding TT-seq time points matched (for 15 and 30 minutes, respectively), however no TT-seq data from 60-minute Jurkat stimulated cells was available, and we therefore colored the 60 min NASC-seq data with the closest available time point (30-min TT-seq data), as stated in the figure legend for Fig 3.